# A Proposal for a Forest Digital Twin Framework and Its Perspectives

**Luca Buonocore** [1,*] 📵, **Jim Yates** [1,2,3] **and Riccardo Valentini** [1,2,3,4,5] 📵

1   Department for Innovation in Biological, Agro-Food and Forest Systems, University of Tuscia,
    Via San Camillo de Lellis snc, 01100 Viterbo, Italy; jim.yates@unitus.it (J.Y.); rik@unitus.it (R.V.)
2   Forest Ecology Unit, Research and Innovation Centre Fondazione Edmund Mach, Via E. Mach 1,
    38010 San Michele all'Adige, Italy
3   Nature 4.0 BC SRL, 38068 Rovereto, Italy
4   Department of Landscape Design and Sustainable Ecosystems, Agrarian-Technological Institute,
    RUDN University, Miklukho-Maklaya Str., 6, 117198 Moscow, Russia
5   CMCC Foundation, Via Augusto Imperatore, 16, 73100 Lecce, Italy
*   Correspondence: luca.buonocore@studenti.unitus.it

**Abstract:** The increasing importance of forest ecosystems for human society and planetary health is widely recognized, and the advancement of data collection technologies enables new and integrated ways for forest ecosystems monitoring. Therefore, the target of this paper is to propose a framework to design a forest digital twin (FDT) that, by integrating different state variables at both tree and forest levels, creates a virtual copy of the forest. The integration of these data sets could be used for scientific purposes, for reporting the health status of forests, and ultimately for implementing sustainable forest management practices on the basis of the use cases that a specific implementation of the framework would underpin. Achieving such outcomes requires the twinning of single trees as a core element of the FDT by recording the physical and biotic state variables of the tree and of the near environment via real–virtual digital sockets. Following a nested approach, the twinned trees and the related physical and physiological processes are then part of a broader twinning of the entire forest realized by capturing data at forest scale from sources such as remote sensing technologies and flux towers. Ultimately, to unlock the economic value of forest ecosystem services, the FDT should implement a distributed ledger-based on blockchain and smart contracts to ensure the highest transparency, reliability, and thoroughness of the data and the related transactions and to sharpen forest risk management with the final goal to improve the capital flow towards sustainable practices of forest management.

**Keywords:** forest digital twin; ecosystem services; blockchain; IoT; Earth's digital models

## 1. Introduction

The value of Earth's ecosystems for human well-being, the economy, and sustainable development has been a prominent topic in the public debate and in the global political agenda for roughly half a century. The scientific community has both conceptualized and quantified the societal benefits of functioning ecosystems through the concept of ecosystem services (ES) [1,2], and it has identified the areas where the re-design of human activities to remain within the planet's boundaries is more urgent [3]. The role of forests in contributing to ecosystem services is paramount and well-known. For example, forests play a pivotal role as a global carbon sink accounting for c. 45% of terrestrial carbon stocks [4,5]. Mitigation actions on forests represent more than two-thirds of cost-effective Natural Climate Solutions to limit global warming to below 2 °C by 2030 and about half of the low-cost mitigation opportunities [6]. These mitigation actions on forests as a co-benefit would enhance ecosystem services such as biodiversity (alfa, beta, gamma), water (filtration and flood control) soil enrichment, and air filtration. To this aim, forest management is

increasingly required to consider multiple ecosystem services, thus, identifying the right balance among ecological, economic, and socio-cultural factors [7,8].

This paper aims to propose a framework to design a forest digital twin (FDT) to ultimately exploit the technological advancements of data collection at tree level in combination with well-established forest monitoring technologies like remote sensing. Namely, the framework attempts to combine and integrate the monitoring of the state variables of the tree and those of the related surrounding environment with the monitoring of state variables at the forest level. Particular focus toward forest ecosystem services and their monitoring underpins such a design, whereby blockchain technology is proposed as a tool for the implementation of transactions related to such forest ecosystem service assets.

A digital twin is "a virtual representation of real-world entities and processes, synchronized at a specified frequency and fidelity". The design of a digital twin is motivated by a system-specific and user-defined outcome and it relies on physical-virtual integration and historical and real-time data to build present representations and forward-looking models optimizing decision-making and effective action [9]. One key aspect of the digital twin is the "twinning": the synchronization of the virtual and physical states via measurement of the physical entity and the related update of the virtual entity so that both states are "equal" [10]. Digital twin technology is currently widely-used where it occupies a core pillar of the Industry 4.0 revolution, specifically manufacturing [11]. Via technological innovation and gains, it is rapidly entering into new application fields, including Earth's ecosystems [12]. Indeed, the European Commission considers the development of high precision Earth's digital models one of the pillars of the European Green Deal and Digital Strategy and launched in 2021 "Destination Earth" with the goal to deliver the first two digital twins on extreme natural events and climate change adaptation by 2024 [13]. Twinning the Earth's ecosystems is possible because of the advancement of data collection where we have experienced an explosion of the amount of data recorded in different ways, making use of innovations in technologies associated with IoT (Internet of Things) devices, satellites, drones, and microprocessors. Nanotechnologies, superior computational power, 5G wireless broadband connectivity, and semiconductors are the technology enablers of this phenomenon. Consequently, there is an emerging digital stack that is pivotal to proper recording, validation, and access to ecosystems' data. Furthermore, proper recording of data both in terms of volumes and quality will unlock the potential of AI-based solutions [14]. This new data stream and related modeling can not only contribute to the ES estimation [15] but also have the added benefit of being coupled with technologies like blockchain for tracking the ES valued assets.

Blockchain technology allows mutually mistrusting entities to implement value transactions with no need of a central authority while offering tamper-proof and transparent data storage [16] and therefore, although a blockchain network cannot guarantee the end to end correctness of data registered by sources such as IoT devices, it can store smart contracts and trigger actions based on these values [17]. Therefore, autonomous digital entities can be built by implementing smart contracts for sustainable management of natural assets, thus improving ownership, traceability, incentives mechanisms, and governance [18]. Applied to the Forest Digital Twin, the advantage of blockchain technology is not in the evaluation of ecosystem services specifically. Blockchain technology can act, however, as a technological layer to store and transact the calculated asset for a specific ecosystem service (e.g., unique identifier of the amount of C sequestered by a particular stand in one year) without the need of a trusted third party. Furthermore, the technology could provide a tamper-proof full log of the recorded state variables both at tree and forest level values to reconstruct the evolution of the forest ecosystem services.

In this paper, we propose the building blocks of an FDT solution enabling the system of recording, reporting, accounting, and trading of ecosystem assets generated by a forest. The aim is to provide a high-level blueprint to use as a reference for project implementation and to adapt and expand based on project peculiarities. This blueprint aims at ensuring the following benefits:

1. enhance monitoring and verifiability of forest ecosystem services status;
2. reduce the risk posture of the health of forest ecosystem services by implementing early warning mechanisms based on the tracking of state variables at tree and forest levels;
3. increase liquidity of the ecosystem assets and improve the capital flow towards sustainable practices of forest management.

## 2. The Forest Digital Twin: Framework Description

The building blocks of the proposed FDT solution are illustrated in Figure 1 and are detailed in the next sections. We defined the building blocks starting from the literature review conducted by Jones et al. [10] on digital twins systematizing their recurrent characteristics. The conceptual framework first identifies two separated spaces: real and virtual. At the core of the real space stands the tree as the fundamental component of the forest ecosystem. Indeed trees are the building blocks of the observation components to be included in the FDT. The tree is formed by three layers: the first is the tree and near environment physical and biotic state variables layer (e.g., sapflow and sap flux density, air quality), the second includes the real–virtual digital sockets that are the tools to record and digitize the state variables (e.g., spectrometers to capture lights spectral components of the tree canopy, radial growth via dendrometry sensors), the third is the physical and physiological processes layer that the tree and its surrounding environment realize (e.g., biomass production, tree photosynthesis). This model is replicated for multiple trees, reported in the Figure as 1 to $n$. We propose a nested approach putting the forest layers on top of the trees' layers. We define as nested approach the twinning of multiple trees together with the forest entity using a common data modeling language.

Following the same concept of the tree, the forest is formed by the same three layers but operating at a higher spatial level: first, the physical and biotic state variables of the forest and the surrounding environment (e.g., the atmospheric gaseous and energy exchanges), second, the real–virtual sockets capturing and digitizing the key data of forest status (e.g., LiDAR remote sensing for assessing forest disturbance regime), and third, the physical and physiological processes of the forest (e.g., evapotranspiration and hydrological balance). This nested design principle enables to integrate over time and for different spatial scales (from tree to forest) the related physical and physiological processes by using, in an integrated manner, a wide range of technological means for implementing the real–virtual sockets. Furthermore, it considers biotic and abiotic interactions among trees enabling the prediction of the forest not only by a sum of tree behaviors but also from their interactions, taking into account also non-linear effects at forest stand scale. This approach could be important, for example, for a better understanding of drought-induced tree and forest mortality [19]. In fact, although much progress has been made so far to monitor and to model the causes of tree mortality due to extreme events, the current understanding of tree physiological responses to drought and heat still does not enable a realistic estimate of forest mortality episodes under rapid climate change [20]. An FDT can contribute to generating a relevant amount of data to model the ecophysiological mechanisms at different scales influencing the tree response under stress.

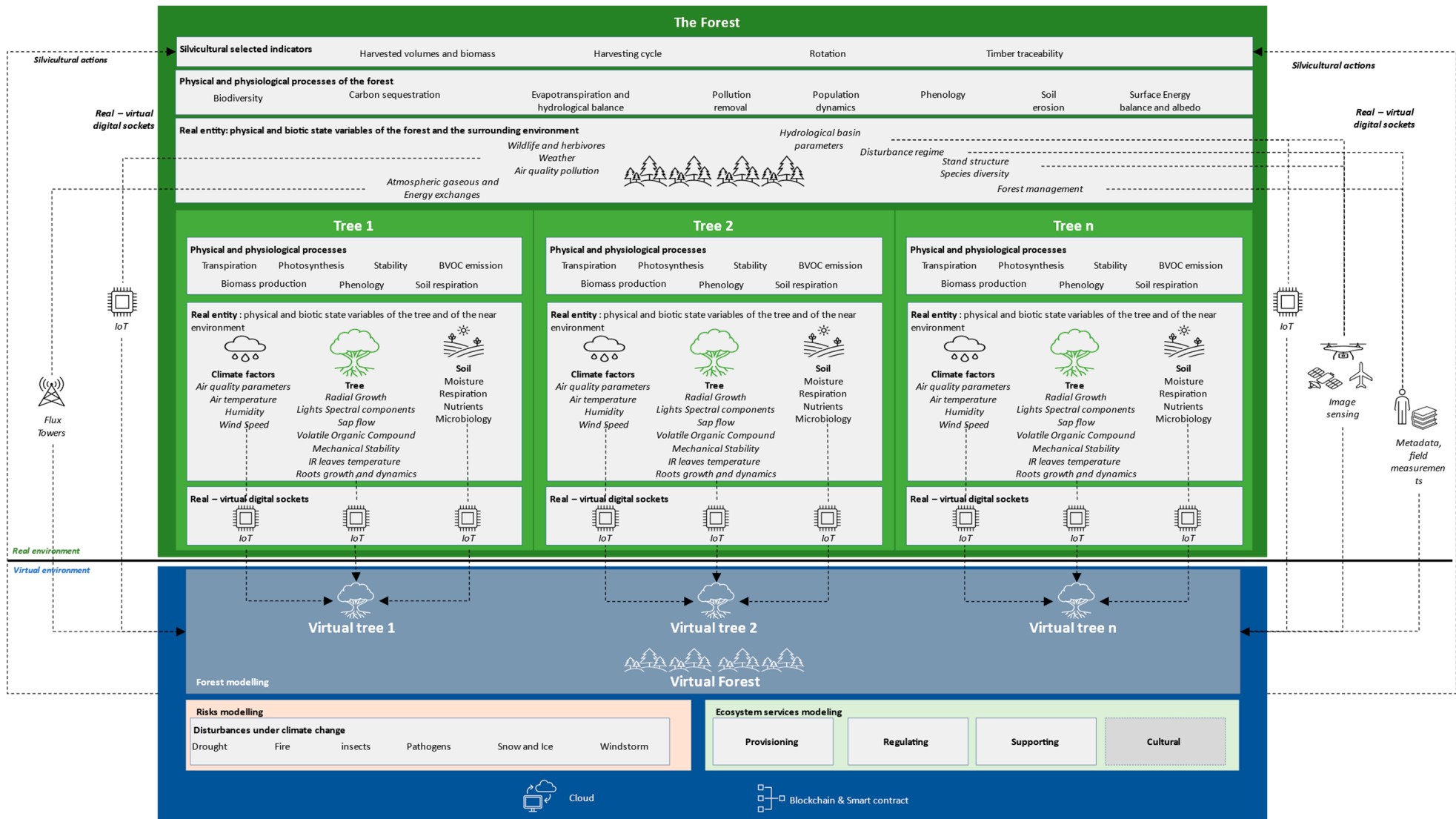

**Figure 1.** Forest digital twin framework.

The real environment at the forest level could ultimately implement a set of indicators to monitor the impact of silvicultural actions. We suggest the following four main indicators:

- Harvested volumes and biomass: whereby routine silvicultural intervention for raw material procurement or biomass removal for successional or sanitation purposes may be fed back into the FDT providing necessary information on past actions to inform future interventions;
- Harvesting cycle: established harvesting cycles, not only as planning tools, may also be embedded where scenarios based on the virtual component of the FDT simulations offers additional information under certain temporal scales and their respective outcomes, e.g., mean annual increment (MAI) stand or forest level, or at tree level average sawlog volume (ASV);
- Rotation: similar to harvest cycles, rotation age information could be continuously assessed and modified under a scenario-based predefined objective or simulated to reflect changes in abiotic and biotic limitation, including changes in timber market conditions;
- Timber traceability: tracing harvest removals (sawlog or biomass) from forest to landing to sawmill with a specific identifier or barcode, which may be tagged as either part of the blockchain ledger or as a specific subset within the FDT to clearly trace raw material forest source and additional repository functions.

For the purpose of this paper, we define the real-virtual digital sockets as the data connections measuring the state variables of the trees and the forest, this includes:

- IoT device: a device or a system of devices equipped with a unique identifier that measures physical and/or biotic parameters that transmit and receive signals over a network. In this context, this classification includes (list no exhaustive): sap flow sensors, diameter growth sensors, sound piezometric sensors, soil moisture sensors, air quality sensors, digital cameras to capture animal movements, weather stations;
- Image sensing technologies: remote sensing (satellite images and light detection and ranging (LiDAR) technology operated by drones and planes) and terrestrial laser scanners;
- Flux towers: measures of gas exchanges (carbon dioxide and water vapor exchange) between the forest ecosystem and the atmosphere;
- Field measurements and metadata: measurements in-situ performed by individuals and usage of additional data sources (e.g., forest inventories).

Symmetrically to the real environment, at the core of the virtual environment stand the virtual entities of trees made by the data captured and the models of the related real physical and physiological processes. At a higher nested level stands the virtual entity of the forest formed by the tree virtual entities plus the data captured at the forest level and the related models of the forest's physical and physiological processes. The virtual space is where the forest models could operate (e.g., species distribution models, individual-based models, dynamic global vegetation models), benefiting from a continuous stream of information [21]. Ultimately, this set of information and models act as an input for the models for risk mitigation and early warnings and for the estimation and management of the forest ecosystem services.

We suggest this nested approach in the virtual environment to address multiple ecosystem services with different levels of granularity including [22]:

- the compositional context (from vegetation association and habitat types to single stand characterization);
- the temporal context (from strategic planning to operations);
- the spatial scale (from regional to single stand);
- the spatial context (for example, to manage the harvest units or wild habitat patterns);
- the decision-making context (from multi-decisions/multi-stakeholder to single stakeholder).

Therefore, to meet this design principle, the simultaneous collection and integration of state variables and process models at both tree and forest levels is a prerequisite the FDT should meet.

Ultimately, we propose to operationalize the virtual environment of the FDT on two main technologies: (1) cloud computing and (2) blockchain and smart contracts. Virtual entities and models of the real environment exist in a virtual environment based on the continuous collection and analysis of data. In this respect, on the basis also of gray literature review, cloud computing provides over the internet an evolving list of services for the set-up of an FDT such as servers, storage and database, networking, connectivity and control services (e.g., for IoT devices), computational scalability, data engineering, and analytics capabilities to process a continuous stream of data from multiple sources [23]. Some cloud providers also offer digital twin capabilities as a platform as a service solution (PaaS) to build relationships based on the digital models of the environment to be modeled [24]. However, in the future, we envisage an increasing use of scalable edge computing technologies which could deploy FDT AI predictions at locale scale (e.g., park management office, timber companies, etc.) thanks to the rapid expansion of GPU multicores parallel computing technologies (e.g., JetsonNano, etc.) with a significant cost reduction and network reliability.

Blockchain and Smart contracts (low-level code scripts running on a blockchain platform [25]) have the advantage of unlocking the economic value of forest ecosystem services. In this respect, the virtual environment of the FDT is suggested to implement an accounting system to ensure the highest transparency, reliability, and thoroughness of the data enabling automatic validation of the data collected at tree level with the data collected at the larger forest scale. Without such a solution, with the growth of data collection and models outputs, recording, reporting, accounting, and trading of forest ecosystem assets will be still affected by potential credibility failures, including double-counting, lack of transparency in the transactions, and limitation in the information access [26,27]. This distributed ledger should act as a trusted source of forest's status and ensure interoperability and integration among different implementations of FDT.

Blockchain and smart contract technology could fit for purpose because it covers all the essential components we identified as required to develop a ledger of the digital forest to fully unlock the value of the forest ecosystem services [28]:

- enable different actors to have access and operate on the data recorded in a decentralized way on the basis of the specific use cases;
- transfer of digital assets and information without requiring a trusted third party: value sharing related to digitized natural assets like a carbon credit or like any value linked to a forest ecosystem service is fundamental, and the potential variety of actors and use cases enabled by the FDT make the establishment of a trusted third party difficult to realize;
- clear data ownership: digital identifiers for forest ecosystem services evaluations (e.g., wood production, carbon sequestered, pollution removal) linked to a trusted ownership mechanism are mandatory for accountability and transparency;
- enable monitoring activities: depending on the use cases, the public sector, the scientific community, and civil society should have real-time access to the transactions. Data reconciliation must be an "off of the shelf" capability to minimize disputes and manual efforts. A full log of transactions must also be available to reconstruct the evolution of the forest ecosystem services.

One key aspect to be considered during the technical design phase of the FDT is to limit the carbon footprint generated by the blockchain platform underpinning the transactions [29]. Therefore, it is required to use a blockchain platform that does not generate an exponential growth of energy consumption to add blocks to the ledger and to increase performance.

Compared to generally used forest management and decision support systems [30], we identify two main advantages that the FDT framework would bring:

- thanks to a cloud-native design, similar to the digital twin implementations for manufacturing, the solution would underpin the collection, storage, and analysis of data recorded in different ways and forms (e.g., IoT, remote sensing, and national forest inventories), creating a new data lake from tree to forest level available for modeling;
- the punctual monitoring at tree level, could enable a better prediction of the impact of forest management actions on ecosystem services. A clear example could be the impact evaluation of biomass removal via harvesting for a stand. Monitoring the state variables of the trees forming the stand via IoT devices would enable the daily collection of data pre and post-harvesting, for example: for radial growth (via growth sensors), canopy health (via spectrometer), tree competition (via GPS), and stem water usage (via transient thermal dissipation probs), thus providing a continuous stream of data for assessing the impact of the forest management activities.

### 2.1. Twinning the Tree: The Physical and Biotic State, the Sockets, and the Processes

To create a digital twin of the tree, we suggest measuring and recording 15 sets of variables clustered by tree (IR leaves temperature, lights spectral components, radial growth, roots growths and dynamics, sap flow, volatile organic, tree mechanical stability), by the soil where the tree stands (soil microbiology, moisture, nutrients, respiration) and, by the climate factors in the near space of the tree (air temperature, air quality parameters, humidity, wind speed). The real–virtual sockets should be made by IoT devices operating in wireless mode to ensure the highest level of resolution and frequency (ideally sub-hour). To our best knowledge, the technological means to implement this type of connection are already available, with the exception of:

- roots growth and dynamics of the tree: expensive, limited to white-light imaging, and often requiring human intervention [31];
- soil nutrients: reduced commercial availability [32];
- soil microbiology: under development for specific use cases [33].

This set of data enables the process-based modeling in semi-real time mode of at least seven physical and physiological processes of the tree and of its surrounding environment: BVOC (biogenic volatile organic compounds) emission, tree biomass production, tree phenology, tree photosynthesis, soil respiration, tree stability, tree transpiration defining therefore the behavior of the virtual entity tree within the virtual forest. Table 1, reports the state variables and the related IoT connections and gaps. Furthermore, it provides a selection of scientific references demonstrating the usage of these variables and IoT connections for the analysis of the physical and physiological processes of the tree and its surrounding environment.

### 2.2. Twinning the Forest: The Physical and Biotic State, the Sockets, and the Processes

To create a digital twin of the forest we suggest measuring and recording state variables related to 9 clusters: air quality and pollution, atmospheric gaseous and energy exchanges, disturbance regime, forest management, hydrological basin parameters, species diversity, stand structure, weather, wildlife and herbivores. The real–virtual sockets could consist of:

- IoT devices for monitoring air quality and pollution, weather, wildlife, and herbivores;
- image sensing technologies for monitoring species diversity and stand structure;
- flux towers to monitor atmospheric gaseous and energy exchanges;
- field measurements and metadata analysis for monitoring forest management activities and tree competitions;
- combinations of these monitoring means for disturbance regime (remote sensing and metadata) and for the hydrological basin (remote sensing and IoT).

**Table 1.** Twinning the tree. The table reports the state variables and the related IoT availability to realize the real-virtual socket. It also indicates the main relevance (✓) of the variables to model the physical and physiological processes for the tree virtual entity, and it provides selected references for each state variable typology.

| Typologies of State Variables | | IoT Availability | Physical and Physiological Processes: Variables Relevance for Process Modeling and Selected References | | | | | | |
|---|---|---|---|---|---|---|---|---|---|
| | | | BVOC Emission | Biomass Production | Phenology | Photosynthesis | Soil Respiration | Stability | Transpiration |
| Tree | Mechanical stability | Available | | | ✓ | | | ✓ | |
| | | | | | | [34,35] | | | |
| | IR leaves temperature | Available | ✓ | | ✓ | ✓ | | | ✓ |
| | | | | | | [36] | | | |
| | Light spectral components | Available | ✓ | ✓ | ✓ | ✓ | | | ✓ |
| | | | | | | [35,37] | | | |
| | Radial growth | Available | | ✓ | ✓ | ✓ | | | ✓ |
| | | | | | | [38] | | | |
| | Roots growth and dynamics | Reduced availability | | ✓ | ✓ | ✓ | ✓ | ✓ | ✓ |
| | | | | | | [31] | | | |
| | Sap flow | Available | | ✓ | ✓ | ✓ | | | ✓ |
| | | | | | | [36,38,39] | | | |
| | Volatile organic | Available | ✓ | | ✓ | ✓ | | | |
| | | | | | | [40,41] | | | |
| Soil | Microbiology | Reduced availability | | ✓ | ✓ | ✓ | ✓ | | ✓ |
| | | | | | | [33,42,43] | | | |
| | Moisture | Available | ✓ | ✓ | ✓ | ✓ | ✓ | ✓ | ✓ |
| | | | | | | [44] | | | |
| | Nutrients | Reduced availability | ✓ | ✓ | ✓ | ✓ | ✓ | | ✓ |
| | | | | | | [32] | | | |
| | Respiration | Available | ✓ | ✓ | ✓ | ✓ | ✓ | | |
| | | | | | | [45,46] | | | |
| Microclimate factors * | | Available | ✓ | ✓ | ✓ | ✓ | ✓ | ✓ | ✓ |
| | | | | | | [36,38,47] | | | |

* Microclimate factors include air temperature, air quality parameters, humidity, and wind speed.

With the exception of the monitoring related to the status of forest management activities and tree competition, to our best knowledge, all other state variables can be gathered via technological means remotely connected, thus enabling a continuous generation of stream of data. This flow of information enables the modeling of at least eight physical and physiological processes of the forest, forming the behavior of the FDT in the virtual space: biodiversity, carbon sequestration, evapotranspiration, phenology, population dynamics, pollution removal, soil erosion, surface energy balance and albedo. Table 2 summarizes the state variables, the real-virtual sockets, the processes, and a selection of scientific references.

### 2.3. Risk Management and Early-Warnings

Risk management and early-warning mechanisms should be a must-have capability of the FDT for two main reasons: first, to support forest risk mitigation actions; second, to provide distinctive information to key forest stakeholders. Notable examples of distinctive information could be:

(1)  the risk of reversals impacting the asset value provided by an ecosystem service shared with a potential buyer (e.g., risk of wind storm hindering the forest's potential of carbon sequestration);

(2)  signals of the forest's health degradation for the scientific community (e.g., increasing episodes of hydraulic failure, carbon starvation, insects, and pathogens).

**Table 2.** Twinning the forest. The table reports the state variables and the related real-virtual socket. It also indicates the main relevance (✓) of the measured variables to model the physical and physiological processes representing the forest virtual entity, and it provides selected references for each state variable typology.

| Typologies of State Variables | Real–Virtual Sockets | Physical and Physiological Processes: Variables Relevance for Process Modeling and Selected References | | | | | | | |
|---|---|---|---|---|---|---|---|---|---|
| | | Carbon Sequestration | Phenology | Population Dynamics | Biodiversity | Pollution Removal | Soil Erosion | Evapotranspiration and Hydrological Balance | Surface Energy Balance and Albedo |
| Air quality and pollution | IoT | | ✓ | ✓ | ✓ | ✓ | | | |
| | | | | | | [48] | | | |
| Atmospheric gaseous and energy exchanges | Flux towers | ✓ | ✓ | ✓ | ✓ | ✓ | | ✓ | ✓ |
| | | | | | | [49,50] | | | |
| Disturbance regime | Immage sensing/metadata | ✓ | ✓ | ✓ | ✓ | ✓ | ✓ | ✓ | ✓ |
| | | | | | | [51,52] | | | |
| Forest management | Field measurements and metadata | ✓ | ✓ | ✓ | ✓ | ✓ | ✓ | ✓ | ✓ |
| | | | | | | [53] | | | |
| Hydrological basin parameters | Remote sensing, IoT | ✓ | ✓ | ✓ | ✓ | | ✓ | ✓ | ✓ |
| | | | | | | [54] | | | |
| Species diversity | Remote sensing | ✓ | ✓ | ✓ | ✓ | ✓ | ✓ | ✓ | ✓ |
| | | | | | | [55,56] | | | |
| Stand structure | Remote sensing, IoT | ✓ | ✓ | ✓ | ✓ | ✓ | ✓ | ✓ | ✓ |
| | | | | | | [57–62] | | | |
| Weather | IoT | ✓ | ✓ | ✓ | ✓ | ✓ | ✓ | ✓ | ✓ |
| | | | | | | [63] | | | |
| Wildlife and herbivores | IoT | ✓ | ✓ | ✓ | ✓ | | ✓ | ✓ | ✓ |
| | | | | | | [64,65] | | | |

Ideally, by adopting common standards for tracking and reporting, multiple implementations of FDT could build a global network like Fluxnet [49], thus, generating a distinctive globally data-driven repository for the scientific community. Furthermore, reliable data on threats affecting forest ecosystems can improve risk awareness and foster the implementation of mitigation actions [66].

The FDT could implement, as required risk management capability, the risk modeling of the disturbances under climate change, including on the basis of the relevance for specific FDT implementation of fire, drought, wind, snow and ice, insects, and pathogens [67]. This risk component is proposed as mandatory not only because it is considered as a key risk factor of forest ecosystem status but also because it is instrumental for the creation of a shared risk monitoring capability among different FDT implementations. It is important to outline that forest disturbances and risks are by nature interlinked [68], and have different temporal and spatial resolutions, therefore, the risk models cannot be built in silos, but they should follow an integrated approach.

### 2.4. Ecosystem Services Evaluation: The Value of Tree Monitoring

Forest-related ecosystem services present a specific suite of opportunities and pitfalls given their inherent complexity. They are dynamic systems changing in space and time with various hierarchical system processes and dependencies interwoven in a complex matrix of both natural succession and anthropogenic induced change. Hence, the importance of dynamic approaches toward the quantification, preservation, and enhancement should be of the utmost importance and much has been achieved in forest ecosystem service quantification and optimization [69,70].

As a design principle, the FDT should provide as distinctive capability the evaluation of forest ecosystem services based on the state variables collected at both tree and forest levels, bridging the gap between static field measurements and autonomous data acquisition. One notable example of the advantage that this capability could bring in this context is the study provided by Klosterman et al. 2014 [71], where the authors used near-surface digital repetition photography and satellite remote sensing to estimate landscape-scale phenological metrics. This study highlights that under specific conditions (heterogeneous land with lower fractional forest cover), late spring phenology estimations derived from satellite remote sensing occur later than the projections from near-surface measurements. Therefore, using just remote sensing data to estimate the phenology of deciduous trees could result in a lower estimation of the annual sums of net productivity for the forest. As such, access to real-time data sources from IoT and simulated responses via a digital twin could possibly harmonize or better refine such estimations with a bottom-up approach.

The monitoring of physical and physiological processes at tree level, reported in paragraph 3.2, enables as a minimum capability the estimation of ecosystem services [38] reported in Table 3, providing a new set of data that can integrate the evaluation of ecosystem services at the forest level. For example, the punctual estimation of carbon removal at tree level can integrate the evaluation of the amount of carbon retained in a forest ecosystem for the carbon pool "carbon in aboveground biomass" usually calculated following the methods "gain and loss", or "stock change" [72].

**Table 3.** Ecosystem services evaluation at tree level based on continuous monitoring of the physical and physiological processes.

| Ecosystem Service | Type | Unit | Measure Frequency | Tree/Soil Physical and Physiological Processes Monitored |
|---|---|---|---|---|
| Timber production | Provisioning | $m^3$ volume growth | yearly on a daily stream of data | Biomass production |
| C removal | Regulating | Kg C sequestered and stored | yearly on a daily stream of data | Soil respiration |
| Particulate absorption on tree canopy | Regulating | $g/m^2$ | daily | Phenology |
| Gaseous pollutants removal on tree canopy | Regulating | $g/m^2$ | daily | Phenology |
| Water runoff | Regulating | % based on indirect LAI (leaf area index) | daily | Phenology |
| Water runoff | Regulating | L/hour | daily | Transpiration |
| Water runoff | Regulating | Soil volumetric water content | daily | Soil moisture |
| Energy balance regulation | Regulating | $W/m^2$ | daily | Transpiration |

Other physical and physiological processes monitored at tree level can also play the role of contributing indicators for the estimation of other ecosystem services like the monitoring of tree BVOC emission and photosynthesis for pollination ecosystem service.

*2.5. Putting All Together: An End-to-End Theoretical Application of a Forest Digital Twin*

The scenario reported in Figure 2 describes as an example of end to end flow a forest manager aiming at generating credits for the ecosystem services from the forest (e.g., carbon sequestration under the form of carbon credit) to sell them to a credit market (e.g., the voluntary carbon trading market). An important element of this framework is the overcoming of using a third party for certification, which is usually an expensive and complicated burden for the seller.

The scenario is presented in three distinctive phases: (1) "before the trade," (2) "trade and settlement," (3) "post-settlement and retirement" [73]. The scenario reports the interaction of multiple stakeholders and the role of the FDT to implement the transactions among stakeholders via a set of smart contracts overcoming the need for a trusted third party. As a first step, the forest manager defines the state variables of trees and forests and identifies and implements the physical–virtual digital sockets that will realize the twinning. In this phase, the forest manager should also define the spatial scale and the compositional, temporal, and decision-making contexts [22]. For the sake of this, the FDT should provide interfaces to ingest this information from the forest manager. Then, the forest manager ensures an account on a standard-setter's registry for the related credit and the distributed ledger solution of the FDT. The FDT should provide the possibility to connect and share information with standard setters as tradable credits usually adhere to standards for verification. Notable examples for the case of carbon credits are the Gold Standard, Verra's VCS Program, American Carbon Registry, and the Climate Action Reserve [74–77].

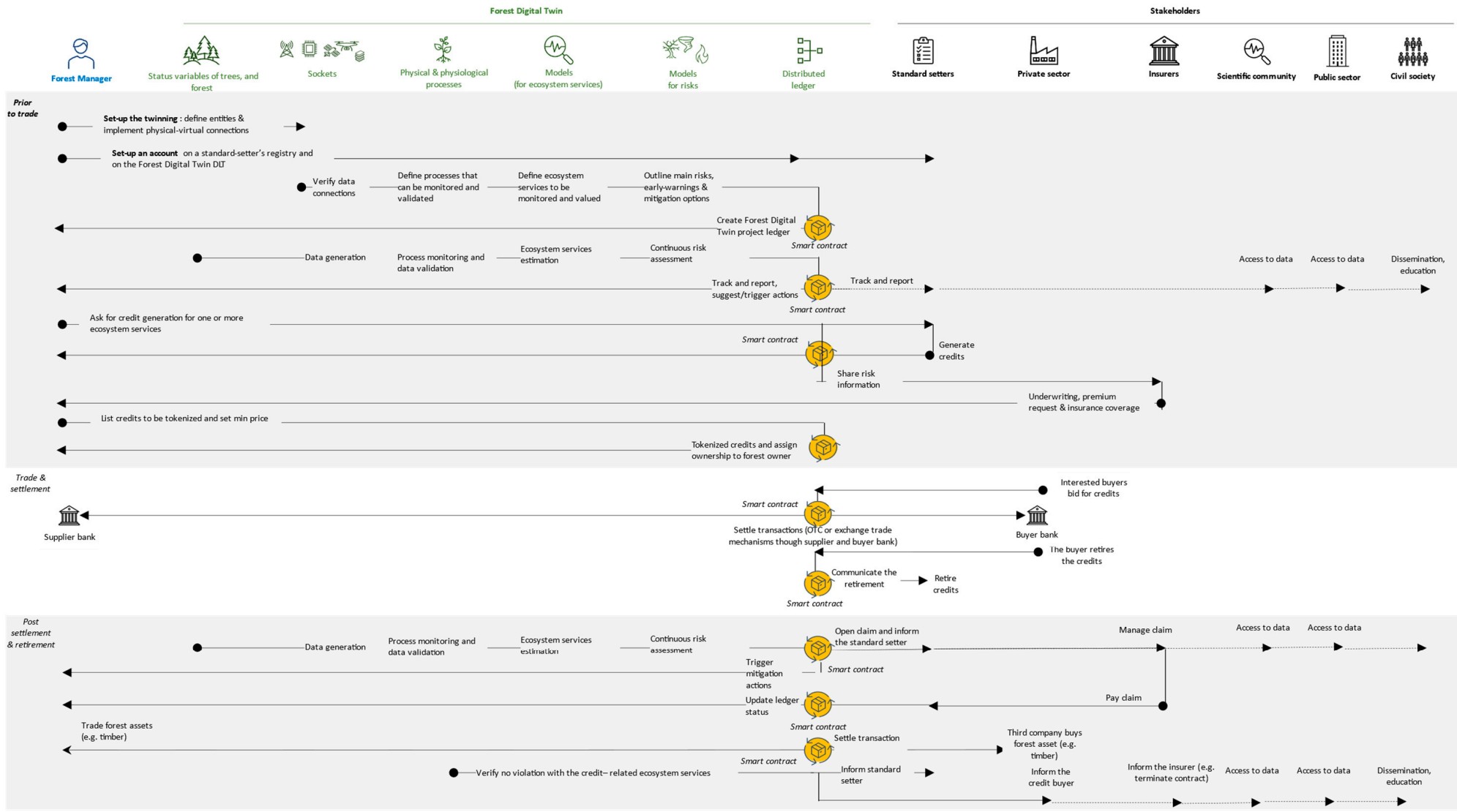

**Figure 2.** An end-to-end theoretical application of a forest digital twin.

The FDT then implements four critical steps: first, it verifies data connections, second, it defines the physical and physiological processes and ecosystem services that can be monitored and valued at tree and forest level, and third, it outlines main risks and sets early-warnings measures. A notable example could be the suggestion of creating a basket of set-aside credits based on the probability of unexpected events (e.g., wildfire, pests) like the Forest Buffer Account for carbon credits [78]. The output of these steps are inputs for the first smart contract recording them in the ledger and notifying the forest manager. Using the digital ledger already in this initial set-up phase ensures that also the forest background data, the project set-up, and the future adjustment (e.g., selection of the physical–virtual connections) from the forest manager can be immutably recorded, thus giving the stakeholders access to the entire track of the project information and reducing information asymmetry. This information can, for example, simplify the evaluation of the eligibility of the project, the estimation of the carbon baseline for proving additionality, and the assessment of permanence. For example, the FDT could provide wildfire risk probability in the project region, thus estimating the expected risk of this reversal undermining the permanence of the sequestered carbon to optimize the buffer account for carbon credits.

After this set-up phase, the twinning can take place. The state variables of the tree, the surrounding environment, and the forest, based on the established connections, can be recorded (data generation) and the related processes modeled and monitored. These data would feed the models defined in the set-up phase to estimate ecosystem services (e.g., biomass production for carbon sequestration) and risks (e.g., wildfire). The second smart contract should track and report progress not only to the standard setters but also to other stakeholders like the scientific community. Two notable examples of triggers for this second smart contract could be (1) the communication to the standard-setter of the carbon sequestered above the established business as usual scenario giving therefore an early indication of the potential credits that the project can generate (2) the early-warning communication to the scientific community of cases of hydraulic failures of trees tracked via IoT devices due to drought and heatwaves.

The FDT could then deploy a third smart contract to support the forest manager in the certification of performance for the specific ecosystem service (e.g., carbon sequestration) and related conversion to credits. For example, the entire data history of the project could be readily available to the validation and verification bodies as part of the standard-setter's procedure to certify the credits. To mitigate the impact related to the risk of unintentional reversals, it could be beneficial to introduce the interaction already at this stage with insurance carriers. Insurers can, in fact, provide compensation for losses in case of reversals, and advanced monitoring solutions combined with machine learning capabilities can improve the attractiveness of nature-based solutions as an asset class where to invest [79].

The fourth smart contract simply implements the tokenization of the carbon credits (the asset) and assigns the ownership to the forest manager, who is able to proceed with the trading process. The tokenization can also implement the split of the asset into subparts in a fully transparent manner, thus enabling the trading at different levels of aggregation of the asset.

Figure 2 also reports the trade and settlement phase where the forest manager, via the fifth smart contract, settles the transaction with buyers, and the standard-setter retires the credits. The FDT solution can bring distinctive value for both the buyer and the seller in this phase because the tradable credit of a specific ecosystem service (e.g., carbon credit) first, is based on a set of data easily auditable and, second, it can be combined with information status and value estimation of other ecosystem services (e.g., air and water filtration, biodiversity) generated by the same forest making, therefore, the credit more attractive for a potential buyer and not just a commodity.

In the post-settlement and retirement phase, the FDT continues to collect data feeding the modes for risk management and ecosystem services. In this phase, the implementation of the smart contract would mitigate the risk of fraud (e.g., data tampering) and will ease

the communication process (e.g., by triggering a notification of loss towards the insurer in case of an event due to a natural catastrophe).

Ultimately, the last smart contract could cover the willingness to trade the forest assets (for example, timber production) on top of the credit generation. Once the forest manager sets the transaction with the buyer, the smart contract could (1) verify that there is no violation with the condition for the credit generation related to the ecosystem asset sold, (2) inform the standard-setter and the buyer, (3) trigger the termination of the insurance contract and (4) inform the public sector and the civil society on the destination of usage of the timber produced triggering, therefore, stakeholder consultations. This last set of transactions, which are fully monitored and auditable, could enable the implementation of innovative solutions like buildings as a global carbon sink [80] to further unlock the value of the forests, thus improving the flow of capital in the forest sector.

## 3. Discussion and Conclusions

The literature on digital twins in a forest systems context appears novel. The potential of using a forest digital twin to integrate different types of data sources to create a multi-purpose digital replica of the forest has not been fully realized nor explored in-depth prior. In fact, novel technological improvements in the systems of recording the state variables of forest ecosystems both at tree and forest levels and the availability of quickly scalable cloud services like servers, storage and database, networking, and connectivity enable the possibility to look at forest ecosystems via the integrated analysis of various data sources. These data sources have different spatial, temporal, and dimensional properties, notably associated with a method of data capture and synthesis. For example, satellite images with varying resolutions and time capture periods, temporal changes in forest carbon flux recorded by flux tower systems, and real-time monitoring of tree sap flow. Not to mention data from national forest inventories and newer techniques for capturing forest information such as mobile LiDAR scanner and subsequent data obtained in field measurements.

The fundamental hypothesis behind the FDT is that it is not only a repository for information gained by the capture and integration of real and physical environment data, but that it represents a novel data-driven systems approach, modulated with the integration of currently emerging sensor applications and connectivity via IoT. This IoT connectivity enables the integration of ecophysiological processes at tree level with the detailed forest structures data sets, thus extending in both spatial and temporal domains forest monitoring capabilities.

In this study, we suggested a coherent framework to integrate these data sources starting from the concept of Digital Twin already used in other industries, namely manufacturing. The FDT framework is open by definition, meaning that it could serve as a basis to build applications from achieving pure scientific purposes to implementing sustainable forest management practices. Regardless of the purpose of the application, we propose some key design principles that should be considered; (1) twinning the tree and the related surrounding environment as the fundamental unit of the forest—this design principle could ensure the highest data granularity by having a stream of information on the physical and physiological processes of trees forming the forest. (2) Integrate the data from the trees with data sources at stand and forest level—this would unlock the possibility to cross-validate and enhance the results of models applied at a forest level. (3) Assess the risk posture of the forest and define how the data collected can provide early warnings on the major threats the forest faces. Forest main disturbances under climate changes should always be assessed. (4) Define the ecosystem services to be monitored and use the data at tree level to support the estimation of these ecosystem services at the forest level. (5) Consider the usage of blockchain technology to store the data captured and to enable economic transactions on the ecosystem services monitored.

The main limitation of the study at this stage is the lack of implementation of the framework on real applications. The end-to-end testing of FDT concept is paramount to validate and refine the model based on the results of concrete use cases. As reported in Tables 1 and 2, applications monitoring selected state variables at both tree and forest

levels are available, but to our best knowledge, an attempt to combine multiple monitoring applications at different scales having the recording of the state variables of the tree as the core entity of the solution is currently missing. We, therefore, suggest as an immediate next research direction the implementation of the framework on a use case to prove the following key hypothesis: (1) the recording of the state variables at tree level can enrich and cross-validate the monitoring of the physical and physiological processes at forest level (2) the measures of ecosystem services can be stored as an asset on a blockchain network with a tree-level granularity without the need of a trusted third party thus providing an accessible and tamper-proof full log of measures. The use case should have a limited and well-defined scope focusing therefore on a specific ecosystem service like carbon sequestration. Once this use case proves to be successful, the model should be tested on both other ecosystem services and framework components (e.g., monitoring the impact of forest management actions, risk simulation, and early warnings). The second main research direction should focus on the standardization of data sharing protocols among different digital twin implementations (including entities attributes and data sources). This aspect was beyond the scope of the article, but it represents a limitation of FDT model at this stage because the adoption of digital twin solutions applied to Earth's ecosystems is expected to grow over the next years and the proliferation of protocols and governance practices also linked to the multiple data sources can generate technical debt, interoperability issues, and governance obstacles thus reducing the value that this type of solution applied to forests would generate at a local and global scale.

**Author Contributions:** Conceptualization, L.B., J.Y. and R.V.; methodology, L.B.; writing—original draft preparation, L.B.; writing—review and editing, L.B, J.Y. and R.V.; visualization, L.B.; supervision, R.V. All authors have read and agreed to the published version of the manuscript.

**Funding:** R.V. is supported by the Russian Scientific Foundation (RSF) grant No. 19-77-300-12.

**Data Availability Statement:** Not applicable.

**Acknowledgments:** We thank three anonymous reviewers for their valuable comments.

**Conflicts of Interest:** The funders had no role in the design of the study; in the collection, analyses, or interpretation of data; in the writing of the manuscript, or in the decision to publish the results.

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
