# Peer review of "A Proposal for a Forest Digital Twin Framework and Its Perspectives"

_forests, doi:10.3390/f13040498_

Round 1
Reviewer 1 Report
In the paper entitled “Towards the management of forest ecosystem services through a Forest Digital Twin”, Buonocore et al. proposed a framework to design a Forest Digital Twin (FDT) for forest management and monitoring. Certainly this topic falls on the scope of FORESTS. However, I am not sure whether the topic might be of interest to the readers. The major differences between FDT and general forest management systems are not clear. Compared to generally used forest management systems, the advantages of the newly proposed framework should be highlighted. On the other hand, the validity of FDT is unclear because of the general description of the newly proposed framework without any application or any case study. In addition, I think the proposed FDT should also be carefully evaluated or validated before publication.
Minor suggestions:
- Page2, line 68: definition of IoT;
- Page 11, line 325-326: the image is blurred.
Reviewer 2 Report
This manuscript deals with designing of a digital twin decision support system applied to forest management and monitoring issues. The aims and objectives of the paper fit in the editorial policy of this journal. My overall recommendation is that the manuscript does not deserve publication in its current form in Forests, although the topic is original and novel. My comments are listed as follows:
• General Comment: The manuscript needs more explanation about critical questions: the advantage of this new decision support system compared with other models and their potential costs. Why is there a need in the forestry field to use this technology? Besides, the ES are not fully represented in this model. For example, nothing is said about how the valuation of the ES can be made. Besides, the main target (l. 47-48) is quite general. It needs more explanation.
• Introduction: It is not clear the real utility of blockchain technology in aspects related to ES. For example, their valuation. On the other hand, the specific objective nº2 needs more explanation.
• Material and Methods Section needs to be rewritten. It is unclear why this bibliographic review has been done, why the authors have chosen Google Scholar (“grey literature”), the specific searches, and their results. It is hard to understand that the model “appears” only with the review, without other starting points. Some hypotheses need to be clarified in this Section. Finally, in the Results Section there are issues not defined in a previous Methods Section (i.e., “nested approach”; “smart contracts”).
• Results: Figure 1 (and maybe Figure 2) needs an improvement (it is not fully readable). L 124-125: I miss a layer associated with forest management issues (Men’s actions in the forests are almost excluded in the methodology proposed). There are a lot of physical and biotic measurements, but indicators associated with men’s actions are not introduced. L. 195. I do not understand the utility of blockchain when nothing is said about how the different ES are valued (and how often). L. 209: Who are the participants? Stakeholders? How are their preferences computed? L. 216: It is assumed any ES can be tradable in a perfect market using specific credits. This is not a result; it is a hypothesis and must be corroborated. Moreover, I do not understand the meaning of the sentence included in l. 254. Why is it so important the carbon footprint of the blockchain proposed? Regarding carbon, the carbon balance depends on the harvests proposed. Nothing is said about this question at the forest level (l. 272). Regarding risk management, I do not understand some sentences (l. 320-305), and nothing is said about the risks in, for example, timber market products
• I miss a Discussion Section, and the Conclusions Section must be rewritten. For example, the last paragraph (about standardization) is not deduced from the results obtained.
Minor comments:
• Explain BVOC acronym.
Reviewer 3 Report
The paper deals with quite a novel topic for forest science, namely using the modern blockchain and digital twins technologies to improve the quality of forest management. This is the strong side of the paper, as such an approach is rarely used both in practice and academic literature.
I would also mention a very right decision to analyze the so-called gray literature to discover the field of study. This is indeed important for topics with a frontier scope.
The detail of figures aimed to show the models of forest digital twin framework and management of ecosystem services in a very comprehensive way is admirable, but this is also a shortcoming, as these figures are not readable. If possible, please consider splitting them into some more detailed figures. The aim of the paper was to “provide a high-level blueprint to use as a reference for project implementation and to adapt and expand based on project peculiarities”, so it is evident that all the material must be very concise and understandable for a broad audience.
Some wrong punctuation can also be found at line 101. The case of points' first words on lines 86—91 is uneven.
Author Response
Dear Editor and Reviewer,
we wish to thank you for the comments and feedback on the manuscript.
To improve the readability, we enlarged both the Figures and put them in landscape format. We also saved them in VSG format to ensure that the zooming does not generate degradation in the resolution.
On the presentation of the results, we also sharpen the conclusion section to make the potential benefits of this framework and the potential outlook crisper.
Round 2
Reviewer 1 Report
This is a revised manuscript. While the scientific contribution of the paper is still ambiguous, there was a significant improvement in the methodology description. Without any applications, I think it is difficult to estimate the valuable of the proposed framework. In addition, there are still several errors in the revised manuscript. I suggest carefully revising it before submitting it to the journal. For example:
- 2-3: Title and the author names are mixed;
- 104;
- 327-329.
Author Response
Dear reviewer,
we thank you for the positive feedback on the improvement we achieved with the first review related to the methodology description. We carefully reviewed the manuscript, and we corrected the points you indicated. Furthermore, we worked to improve the manuscript by implementing the following main changes:
- Introduction (Line 83 – 86): we outlined that new data sources can be used for the estimation of the ecosystem services so that there is a stronger logical link among paragraphs in the introduction and we added a selected reference;
- We removed the word “results” starting directly with the framework description being the scope of the article the presentation of a theoretical framework. In this way, we match your valuable consideration of being the question “Are the results clearly presented?” “not applicable”. We acknowledge that, in the case of theoretical paper, the lack of a dedicated results section is also a common practice for the journal (e.g. the article by Baskent published on “Forests” in 2020 “A Framework for Characterizing and Regulating Ecosystem Services in a Management Planning Context”)
- In the conclusion we outlined the importance of testing and validating the framework on real applications as a first next step. For example, in the section where we describe the research directions we report (L525-7) "Future research directions can be identified as: firstly, the implementation of the framework on real applications to test, validate and refine the model based on the results of real case studies” and we outlined also that the lessons learned from real case studies will be pivotal also for other research directions like the data standardization
kind regards
Reviewer 2 Report
First of all, I must congratulate the authors for the changes made and, above all, their responses to my comments. I think that responding to some of them was not an easy task, and they have done it in a reliable and, at the same time, constructive way.
The original manuscript dealt with an innovative issue, and with the improvements made, but keep in mind some minor issues:
1) Delete line 104 (Besides, in my view, the word "Results" can be omitted in this manuscript.
2) line 234. It appears this sentence "Click or tap here to enter text." Please, remove this typo.
3) line 328: the same typo.
Best regards
Author Response
Dear Reviewer,
We thank you for the positive feedback on the new version of the manuscript. We are glad we were able to address your points.
In the new version of the manuscript, we implemented all the improvements in the text you mentioned, including deleting the “results” word.
Kind regards